# Association between Polymorphism in the Janus Kinase 2 (*JAK2*) Gene and Selected Performance Traits in Cattle and Sheep

**DOI:** 10.3390/ani13152470

**Published:** 2023-07-31

**Authors:** Nicola Oster, Małgorzata Anna Szewczuk, Sławomir Zych, Tomasz Stankiewicz, Barbara Błaszczyk, Marta Wieczorek-Dąbrowska

**Affiliations:** 1Department of Monogastric Animal Science, Faculty of Biotechnology and Animal Husbandry, West Pommeranian University of Technology in Szczecin, 29 Klemensa Janickiego, 71-270 Szczecin, Poland; nicola.oster@zut.edu.pl; 2Laboratory of Chromatography and Mass Spectroscopy, Faculty of Biotechnology and Animal Husbandry, West Pomeranian University of Technology in Szczecin, 29 Klemensa Janickiego, 71-270 Szczecin, Poland; 3Department of Animal Reproduction Biotechnology and Environmental Hygiene, Faculty of Biotechnology and Animal Husbandry, West Pomeranian University of Technology in Szczecin, 29 Klemensa Janickiego, 71-270 Szczecin, Poland; tomasz.stankiewicz@zut.edu.pl (T.S.); bblaszczyk@zut.edu.pl (B.B.); 4National Research Institute of Animal Production, Kraków, Experimental Department, Kołbacz, 1 Warcisława Street, 74-106 Stare Czarnowo, Poland

**Keywords:** ruminants, *JAK2*, tyrosine kinase, reproductive performance, growth performance

## Abstract

**Simple Summary:**

An important element of livestock breeding is the selection that leads to the consolidation and improvement of the traits with utility value. The use of single-nucleotide polymorphism (SNP) markers offers the opportunity to select the appropriate genotype linked to the phenotype, which manifests itself in significant breeding benefits. Polymorphic sites within the genes coding the somatotropic axis are directly and indirectly associated with the phenotype, particularly concerning the production properties of growth and carcass as well as reproductive function. This study analyzed the effect of selected polymorphisms in the Janus Kinase 2 (*JAK2*) gene on performance characteristics of cattle and sheep. The impact of the mutations on the development of selected traits, which are important from the economic standpoint, was evaluated. Such studies can be utilized to enhance the growth characteristics of cattle and sheep, thereby facilitating the development of more profitable and sustainable breeding methods.

**Abstract:**

The Janus Kinase 2 (JAK2) tyrosine kinase is an essential component of signal transduction of the class II cytokine receptors, including the growth hormone receptor. Therefore, it may play a crucial role in the signaling pathway of the somatotropic axis, which influences growth, development, and reproductive traits in ruminants. For this purpose, for three breeds of cattle (Hereford, Angus, and Limousin; a total of 781 individuals), two polymorphic sites located in exon 16 (rs210148032; p.Ile704Val, within pseudokinase (JH2)) and exon 23 (silent mutation rs211067160, within JH1 kinase domain) were analyzed. For two breeds of sheep (Pomeranian and Suffolk; 333 individuals in total), two polymorphic sites in exon 6 (rs160146162 and rs160146160; encoding the FERM domain) and one polymorphic site in exon 24 of the *JAK2* gene (rs160146116; JH1 kinase domain) were genotyped. In our study, the associations examined for cattle were inconclusive. However, Hereford and Limousin cattle with genotypes *AA* (e16/*Rsa*I) and *AA* (e23/*Hae*III) tended to have the highest body weight and better daily gains (*p* ≤ 0.05). No clear tendency was observed in the selected reproductive traits. In the case of sheep, regardless of breed, individuals with the *AA* (e6/*Ear*I), *GG* (e6/seq), and *AA* (e24/*Hpy*188III) genotypes had the highest body weights and daily gains in the study periods (*p* ≤ 0.01). The same individuals in the Pomeranian breed also had better fertility and lamb survival (*p* ≤ 0.01). To the best of our knowledge, these are the first association studies for all these polymorphic sites. Single-nucleotide polymorphisms in the *JAK2* gene can serve as genetic markers for growth and selected reproductive traits in ruminants given that they are further investigated in subsequent populations and analyzed using haplotype and/or combined genotype systems.

## 1. Introduction

Genomic selection using single-nucleotide polymorphisms (SNP) is revolutionizing livestock breeding. The primary advantage of genomic selection, in comparison to traditional pedigree-based methods, is the ability to obtain individuals with the desired phenotype and performance traits in the initial generation of selection. There are many studies confirming the effectiveness of the above-mentioned selection. Furthermore, a strong correlation between phenotypic data and predicted genomic values has been consistently demonstrated [1,2]. 

The biological foundation underlying the development of livestock performance characteristics is rooted in the elements comprising the somatotropic axis [3]. The somatotropic axis in the postnatal life stimulates cell hyperplasia and hypertrophy, thus contributing to the further growth of organs and tissues [4]. Proper growth requires the growth hormone (GH) secreted by the pituitary gland to bind to its specific receptor and activate a complex signaling cascade. The GH receptor (GHR) is initially produced as a dimer and then transported unliganded to the cell surface. Binding of GH to the GHR dimer results in a conformational change of the dimer, activation of intracellular Janus Kinase 2 (JAK2), and phosphorylation of the signal transducer and activator of transcription 5B (STAT5B). The phosphorylated STAT5B dimers are then translocated into the nucleus, where they transcriptionally activate various genes, including insulin-like growth factor 1 (*IGF1*), *IGF* binding protein 3, and the acid-labile subunit (*ALS*) [5]. The presence of two JAK2 molecules results in the transphosphorylation and activation of one JAK2 by the other, which in turn leads to the phosphorylation of downstream GHR tyrosine residues. Phosphorylated and dimerized GHR triggers a multiple-signal cascade translocated to the nucleus, where STATs or other non-receptor tyrosine kinases bind to a specific site in the DNA sequence, activating target genes [6].

The primary structure of JAK2 kinase consists of seven homologous domains (Figure 1), designated JH1–JH7, from the carboxyl terminus to the amino terminus [7]. The C-terminal kinase (JH1) domain is responsible for the correct enzymatic activity, while the JH2 domain (pseudokinase) acts as an inhibitor and regulator of the aforementioned JH1 domain. The SH2 homologous region (JH3–JH4 domains) is located in the center of JAK2 proteins, binding directly to JAK-interacting proteins. The N-terminal part of the JAK is built by the FERM domain (JH5–JH7; F for 4.1 protein, E for ezrin, R for radixin, and M for moesin). The FERM domain is highly conserved among many cellular proteins and is involved in the localization of JAKs in relation to the cell membrane and cytoskeleton–including binding to target receptors, e.g., GHR [8]. Constitutive JAK2 activity can occur in various neoplastic processes [9]. Alternatively, JAK2 inhibition may be associated with additional regulatory proteins. A typical JAK kinase is a relatively large protein (more than 1100 amino acids) with a molecular weight of 120–140 kDa. Depending on the species, the genes encoding the individual kinases are located on different chromosomes [10].

The cytokine-activated kinase (JAK) and signal transducer and activator of transcription (STAT) signaling pathway serves as a crucial link between proteins within the cell, directly influencing processes such as cell proliferation, apoptosis, mammary gland development, lactation, and immune responses. This pathway plays a vital role in transmitting information from cell surface receptors to the cell nucleus, ultimately regulating gene expression through transcription [11]. 

There are many studies confirming the influence of Janus Kinase 2 on the performance characteristics of dairy cows. Among other things, it has been shown that the JAK-STAT pathway regulates lactation and that PI3K/Akt within the JAK-STAT pathway is overexpressed in lactating cows [12]. Analysis of gene deletions in mice has documented the important role of JAK-STAT signaling in lactation and mammary gland development [13]. Moreover, an important role of the JAK-STAT pathway in blood cell differentiation and casein gene regulation during milk production has been documented [14]. Prolactin also uses JAK-STAT signaling and regulates lactation and reproduction in mammals [15]. In the research conducted by Szewczuk [16], a clear relationship was found between *JAK2*/*e20*/*Rsa*I polymorphism and milk-yield traits. The SNP was associated with higher milk, protein, and fat yield. With regard to growth and reproductive traits, there is a lack of literature describing the influence of the *JAK2* gene on, among other things, body weight and daily gains at different periods of the animal’s life as well as fertility, prolificacy, or calving interval. The only studies in this area are the results presented by Padzik and Szewczuk [17], which also involved the bovine *JAK2*/e20/*Rsa*I polymorphic site. In contrast, with regard to the ovine JAK2 gene, a preliminary study by Padzik et al. [18] only focused on allele and genotype frequencies and did not consider animal performance traits.

The aim of the present study was to analyze the possible relationship between polymorphisms in the Janus Kinase 2 (*JAK2*) gene and selected growth and reproductive traits in cattle and sheep. Attention was primarily focused on gene fragments encoding crucial domains that determine JAK2 kinase activity.

## 2. Material and Methods

### 2.1. Animals

A total of 781 blood samples were collected from three breeds of cattle, including Hereford (*n* = 276), Angus (*n* = 345), and Limousin (*n* = 160). In regard to sheep, a total of 333 blood samples of two breeds were collected (138 Pomeranian and 195 Suffolk sheep). All animals were kept on a single farm located in West Pomeranian Province, Poland. All animals were kept in a pasture and indoor system. Feeding was carried out in accordance with the standards accepted for these species [19], based on grass and other roughage and concentrates depending on the season. The animals had constant access to water and salt licks.

### 2.2. DNA Isolation

DNA was extracted using a MasterPure DNA Purification Kit Version II (Epicentre Technologies, Madison, WI, USA) from 3 mL of blood collected into tubes containing EDTA (IMPROVACUTER^®^ K3 EDTA, Guangzhou, China). Briefly, this kit uses a rapid desalting process (red and white cell lysis solutions followed by protein precipitation solution) to remove contaminating macromolecules and avoid toxic organic solvents. Rinsing twice with 70% ethanol and resuspending the DNA in TE buffer (10 mM Tris-HCl (pH 7.5), 1 mM EDTA) allows samples to be stored for many months at <−20 °C.

### 2.3. Sequencing

Three fragments of the bovine *JAK2* gene (covering whole exons 3, 16, and 23) and four fragments of the ovine *JAK2* gene (exons 6, 12, 13, and 24) were selected for sequencing (Appendix A). Each of the selected exons potentially contained a minimum of two known polymorphic sites (according to Ensembl–the ARS-UCD1.2 assembly (cow) and the Oar_rambouillet_v1.0 assembly (sheep)). For each exon, 10 pooled samples consisting of 10 individuals were sequenced (Genomed, Warsaw, Poland). Fluorograms were analyzed using Chromas ver. 2.6.6. (Technelysium Pty Ltd., Sydney, Australia). Only the polymorphic sites that had three or more heterozygote-specific results after sequencing of the pooled samples were selected for further genotyping. An additional consideration was the type of mutation detected (order of importance: missense, synonymous, intronic), and all of the detected SNPs had to be present within all breeds under analysis.

### 2.4. Bovine JAK2 Gene Polymorphism

Because the most used commercial enzymes were not available for the native sequence, genotyping of both polymorphisms detected within the three breeds under study was carried out using a polymerase chain reaction–artificially created restriction site (PCR-ACRS) method (Table 1). Primers were designed using Primer3web ver. 4.1.0 online software (https://primer3.ut.ee/) (accessed on 22 July 2021), and correct enzyme digestions after the introduction of a single-nucleotide mismatch were confirmed online by WebCutter v. 2.0 (http://heimanlab.com/cut2.html) (accessed on 22 July 2021) and NEBcutter V2.0 (http://nc2.neb.com/NEBcutter2/) (accessed on 22 July 2021).

### 2.5. Ovine JAK2 Gene Polymorphism

After analysis of the sequencing data, two polymorphic sites in the ovine *JAK2* gene were selected and genotyped using a polymerase chain reaction–restriction fragment length polymorphism (PCR-RFLP) method. Unfortunately, it was not possible to select an optimal methodology for the third SNP (rs160146160). Therefore, all samples were individually sequenced (Genomed, Warsaw, Poland) (Table 2).

### 2.6. PCR Conditions

The PCR reaction was carried out with a total volume of 25 μL of the solution mixture consisting of 12.5 μL 2X PCR Master Mix (Thermo Scientific, ABO, Gdansk, Poland), 0.1 μL of primers (each at concentration of 10 pmol/mL), 2 μL of genomic DNA, and nuclease-free water up to 25 μL. Amplification was performed using Biometra TGradient thermal Cycler (Analytik, Jena, Germany). The first run was at 95 °C for 5 min, followed by subsequent 33 cycles of 95 °C × 30 s, 60 °C × 45 s (all primers), and 72 °C × 45 s and the last run at 72 °C for 5 min (final elongation). The yield and specificity of PCR products were evaluated by electrophoresis in 2% agarose gels (BASICA LE GQT, Prona, ABO, Gdansk, Poland) with ethidium bromide, viewed under UV light and scored in a gel documentation system.

### 2.7. Digestion

In regard to the first polymorphism within the bovine *JAK2* gene (rs210148032), amplicons (211 bp) were digested with 5U of the *Rsa*I restriction enzyme (37 °C/3 h; MBI Fermentas, ABO, Gdansk, Poland), which recognized one GT↓AC sequence within the PCR product. The second bovine polymorphic site (rs211067160) was also characterized by an amplicon of 211 bp. (However, it relates to a different region of the gene; Table 1) In this case, the *Hae*III restriction enzyme recognized one polymorphic site containing this SNP (5 U at 37 °C/3 h; MBI Fermentas, ABO, Gdansk, Poland). A polymorphism within the 280 bp fragment of exon 6 of the ovine *JAK2* gene was detected using the *Ear*I restrictase (also 5 U, 37 °C/3 h; MBI Fermentas, ABO, Gdansk, Poland). The ovine variant within exon 24 was differentiated using the *Hpy*188III restriction endonuclease (5 U at 37 °C/3 h; New England BioLabs Inc., Ipswich, MA, USA). Electrophoretic separation and visualization were identical to that described above.

### 2.8. Statistical Analysis

Analyses of the association between genotype and selected growth and reproductive traits were carried out on the basis of data obtained from the breeding records of individual herds maintained by the Polish Union of Meat Cattle Breeders and Producers and the Polish Union of Sheep Farmers. For cattle, the following traits were analyzed: birth weight (BWT), weaning weight adjusted to 210 days of age (WWT_210_), average daily gain from birth to weaning (ADG), age and body weight at first calving, and calving interval. Traits studied in sheep included body weight at 2, 30, and 56 days of age; average daily gain from 2 to 56 and 30 to 56 days of age; and body weight at mating. Additionally, fertility, prolificacy, and lamb survival were analyzed.

Statistical analysis was performed using the STATISTICA software (13.3 PL software package, Statsoft Inc., Kraków, Poland, 2020). The differences between particular genotypes were evaluated with Duncan’s test. Statistical calculations were performed using a general linear model (GLM).

The following statistical models were used:(Growth traits—cattle: BWT, ADG, and WWT_210_; sheep: BW2D, BW30D, BW56D, BWM, ADG_2–56_, and ADG_30–56_)
(1)Y_ijklm_ = μ + G_i_ +s_j_ + BYS_k_ + [LS_l_] + e_ijklm_where Y_ijklm_ is the analyzed trait; μ the overall mean; G_i_ the fixed effect of *JAK2* genotype (i = 1, …3); s_j_ the random effect of sire (Hereford j = 1, …31; Angus j = 1, …39; Limousin j = 1, …24; Pomeranian j = 1, …3; Suffolk j = 1, …5); BYS_k_ the fixed effect of birth year/season (Hereford k = 1, …20; Angus k = 1, …20; Limousin k = 1, …20; Pomeranian k = 1, …16; Suffolk k = 1, …16); [LS_l_] the fixed effect of litter size of mother (only in sheep: 1 single, 2 twins), and e_ijklm_ the random error. 
(Cattle: age and body weight at first calving as well as calving interval)
(2)Y_ijkl_ = μ + G_i_ + s_j_ +CYS_k_ + e_ijkl_
where Y_ijkl_ is the analyzed trait; μ the overall mean; G_i_ the fixed effect of *JAK2* genotype (i = 1, …3); s_j_ the random effect of sire (Hereford j = 1, …30; Angus j = 1, …39; Limousin j = 1, …21); CYS_k_ the fixed effect of year/season of 1st calving (Hereford k = 1, …22; Angus k = 1, …27; Limousin k = 1, …21); and e_ijkl_ the random error.
(Only sheep: fertility, prolificacy, and lamb survival)
(3)Y_ijkl_ = μ + G_i_ + LS_j_ + LY_k_ + e_ijkl_where Y_ijkl_ is the analyzed trait; μ the overall mean; G_i_ the fixed effect of *JAK2* genotype (i = 1, …3); LS_j_ the fixed effect of litter size (1 single, 2 twins); LY_k_ the fixed effect of lambing year (Pomeranian k = 1, …6; Suffolk k = 1, …8); and e_ijkl_ the random error.

The chi-square test was used to verify whether each population was in Hardy–Weinberg equilibrium.

## 3. Results

### 3.1. Sequencing

After a total of 100 individuals from three breeds of cattle were sequenced, two polymorphic sites were identified in the coding region of the bovine *JAK2* gene (Figure 2). In the case of the sheep *JAK2* gene, three polymorphic sites were previously identified and described by Padzik et al. [17].

### 3.2. Bovine JAK2 Frequencies

A total of 769 individuals from three breeds of cattle were genotyped for the *JAK2*/e16/*Rsa*I polymorphism (Table 3; Appendix A). Regardless of breed, the largest number of individuals was identified with the *AA* genotype (p.Ile704 variant; 58.3–59.4%), followed by heterozygotes (33–34.4%). In contrast, individuals with the *GG* genotype (p.Val704 variant; 7.25–8.41%) were rarely identified. For this reason, the A allele was most commonly observed (75%, regardless of breed). The genotype frequencies of Hereford and Limousin cows were consistent with HWE (*p*-value 0.25 and 0.24, respectively), while the distribution of genotypes in the herd of Angus cows was inconsistent with HWE (*p*-value = 0.03).

More diverse results between the three breeds of cattle in terms of genotype and allele frequencies were obtained for the *JAK2*/e23/*Hae*III polymorphism (Table 3) after testing 759 individuals. In the case of the Hereford breed, as in the previous SNP, the *AA* genotype was most frequently identified (60.9%), followed by heterozygotes (30.6%). Hereford cows with the *GG* genotype were rarely identified (8.5%). This distribution was not consistent with HWE (*p*-value = 0.01). The opposite situation was noted for the Angus breed. The most numerous were cattle with the *GG* genotype (44%) and slightly less often heterozygous individuals (38.3%). Individuals with the *AA* genotype constituted only 17.7% of the population of Angus cows. This distribution was also not consistent with HWE (*p*-value = 0.00). The distribution of genotypes in the herd of Limousin cows was different. Half of the herd consisted of heterozygous individuals (50.3%). There were also numerous cows with the *GG* genotype (30.1%). The *AA* genotype was the least frequently identified (19.6%). The distribution was consistent with HWE (*p*-value = 0.83). The above results were reflected in allele frequencies. The A allele clearly dominated only in the Hereford breed (76.2%) and the G allele in the Angus herd (63.2%). The allele frequency in the Limousin herd was more even (allele A accounted for 44.8% of cases and allele G for 55.2%).

### 3.3. Bovine JAK2/e16/RsaI Associations

In the case of the polymorphism in exon 16 (rs210148032, p.Ile704Val) (Table 4), the birth weight of Hereford and Limousin calves with the *AA* genotype (p.Ile704) was significantly higher than that of the individuals with the *GG* genotype (p.Val704; +1.8 kg and +2.6 kg, respectively; *p* ≤ 0.05). However, for the Angus breed, the opposite results were reported, with *GG* individuals born heavier by 2.2 kg compared to *AA* individuals (*p* ≤ 0.05). A similar situation was noted for the weaning weight (WWT_210_). Hereford and Limousin calves with the *AA* genotype were significantly heavier compared to the *GG* genotypes (+11.1 kg (*p* ≤ 0.05) and +27.2 kg (*p* ≤ 0.01), respectively). For the Angus breed, the differences in WWT_210_ were nonsignificant. In addition, although only for the Hereford breed, there were significant differences in average daily gains (ADG) (*p* ≤ 0.01). Individuals with the frequent *AA* genotype had higher ADG compared to both heterozygotes (+132 g) and individuals with the rare *GG* genotype (+165 g). For the other breeds, the differences in ADG were statistically nonsignificant. The observed trends were maintained until the first calving. The body weight of the cow at the first calving differed significantly depending on the identified genotype. In the case of Hereford, individuals with the *AA* genotype were significantly heavier than both heterozygous individuals (+17.9 kg, *p* ≤ 0.05) and individuals with the rare *GG* genotype (+28.8 kg, *p* ≤ 0.01). In contrast, Limousin cows with the *AA* and *AG* genotypes had an almost identical average body weight at first calving (~610 kg) and were significantly different (*p* ≤ 0.05) from the *GG* genotypes, which had an average body weight 29 kg lower than the other cows.

Some relationships were also noted for selected reproductive traits of cattle, although they were statistically confirmed only for Hereford and Angus breeds. Hereford individuals with the *GG* genotype, despite having the significantly lowest birth weight and body weight in later periods, as described above, had the lowest average age of first calving (~27 months) compared to the other genotypes, with the difference between the *GG* genotype and the heterozygous *AG* genotype being 162 days (i.e., ~5.5 months difference, *p* ≤ 0.05) and between *GG* and the most common *AA* genotype being as much as 226 days (i.e., ~7.5 months difference, *p* ≤ 0.01). In the case of the Angus and Limousin breeds, no significant differences were found because the age of the first calving was similar, occurring at around 33–36 months of age for cows of both breeds. Cows with the *GG* genotype also had a shorter calving interval. Within the Hereford breed, the differences were as much as 100 days compared to cows with the *AA* genotype (*p* ≤ 0.05). Also, heterozygotes in the *JAK2*/e16/*Rsa*I polymorphism for the Hereford breed had a 79-day shorter calving interval compared to cows with the *AA* genotype (*p* ≤ 0.05). The calving interval of Angus cows was even more diverse within the genotypes. As with Hereford, cows with *GG* genotype had a 27-day shorter calving interval compared to cows with *AA* genotype (*p* ≤ 0.01) and 14 days shorter than heterozygous cows (*p* ≤ 0.05). Heterozygotes also had a shorter calving interval by 13 days compared to *AA* cows (*p* ≤ 0.05). In the case of the Limousin breed, there were no significant differences in the calving interval of cows with the different *JAK2*/e16/*Rsa*I genotypes. Irrespective of breed and genotype, all calvings were single and uncomplicated.

### 3.4. Bovine JAK2/e23/HaeIII Associations

Regardless of breed, there was no association between the silent mutation in exon 23 (rs211067160) and birth weight of calves (Table 5). In the case of weaning weight in the Hereford and Limousin populations, individuals with *AA* genotype were heavier by 16.4 kg and 14.3 kg, respectively, compared to individuals with *GG* genotype (*p* ≤ 0.05) and had better average daily weight gains of 154 g (*p* ≤ 0.01) and 91 g, respectively (*p* ≤ 0.05). In addition, heterozygous Hereford cows also had higher ADG than *GG* cows (+112 g; *p* ≤ 0.01). However, Limousin cows with the previously preferred *AA* genotype had 36.8–38.3 kg lower body weight at first calving compared to cows with the other genotypes (*p* ≤ 0.05). For other breeds, no statistical differences in body weight at the first calving were noted.

Similarly to the previously discussed polymorphic site, the age of first calving in Angus and Limousin cows (from 34 to 36 months) did not depend on *JAK2*/e23/*Hae*III genotypes in contrast to Hereford cows, for which, again, individuals with the rare *GG* genotype had significantly earlier age at first calving (~29 months) than *AA* cows (~34 months) (*p* ≤ 0.05). At the same time, cows of this breed, also with the *GG* genotype, had a 90-day shorter calving interval compared to the *AA* genotype (*p* ≤ 0.05). In the population of Limousin cows, the most numerous heterozygotes had a significantly longer calving interval by 90 days than cows with the rarest *AA* genotype (*p* ≤ 0.05). 

For the Angus breed, no statistically significant differences were observed between *JAK2*/e23/*Hae*III genotypes for the growth and reproductive traits. Additionally, it was noted that regardless of breed or genotype, all births were single and occurred without complications or human assistance.

### 3.5. Ovine JAK2 Frequencies

For three polymorphic sites, 138 Pomeranian and 195 Suffolk individuals were genotyped (Table 6; Appendix A).

In the case of the first polymorphic site (*JAK2*/e6/*Ear*I), for the Pomeranian breed, numerous occurrences of individuals of each genotype were noted, with the largest number of individuals being heterozygous (42%), followed by sheep with the *GG* genotype (34.8%), and slightly less often, individuals with *AA* genotype were identified (23.2%). Allele frequencies were similar (A, 44.2%; G, 55.8%). The genotype distribution was consistent with HWE (*p*-value = 0.08). However, in the flock of Suffolk sheep, the *GG* genotype (63%) and the G allele (74%) clearly dominated, and the distribution of genotypes was not consistent with HWE (*p*-value < 0.01). 

In the case of the second polymorphic site (*JAK2*/e6/seq), heterozygotes (POM 42.7% and SUF 56.4%) were the most numerous in both breeds, followed by individuals with the *AA* genotype (39.9% and 25.1%, respectively) and with the *GG* genotype (17.4% and 18.5%, respectively). The A allele occurred more frequently than the G allele (61.2% vs. 53.3%). The distribution of genotypes in both populations was consistent with HWE (*p*-value = 0.24 for the Pomeranian breed and *p*-value = 0.06 for the Suffolk breed). 

The polymorphism in exon 24 (*JAK2*/e24/*Hpy*188III) was also characterized by a slight dominance of heterozygous individuals (POM 48.6% and SUF 65.6%), followed by sheep with the *AA* genotype (31.2% and 25.1%, respectively) and the least frequent being sheep with the *GG* genotype (20.3% and 9.2%, respectively). The frequencies of the A allele were similar (55.4% and 57.8%). The distribution of genotypes in the Pomeranian sheep population was consistent with HWE (*p*-value = 0.84) and inconsistent with HWE in the Suffolk sheep herd (*p*-value < 0.01).

### 3.6. Ovine JAK2 Associations—Growth Traits

At the early stages of animal life, highly significant differences depending on the genotype were observed for most of the analyzed growth performance traits and for each polymorphic site (Table 7). The exception was the subsequent body weight at mating, which was significant only for the Suffolk breed.

Sheep with the *AA* genotype for the *JAK2*/e6/*Ear*I polymorphic site were significantly heavier than the other sheep on day 2 (+0.3–0.7 kg for Pomeranian and +0.7–0.8 kg for Suffolk), on day 30 (+1.2–1.3 kg; only Suffolk), and on the 56th day of life (Pomeranian +1.2–2.2 kg and Suffolk 2.8–3.6 kg) and at mating (+4.1 kg; only Suffolk), especially from the worst-performing individuals with the *GG* genotype (*p* ≤ 0.01) compared to heterozygous sheep and sheep with the *AA* genotype (Pomeranian +16–27 g and Suffolk +40–53 g) between 2 and 56 days of age. For Suffolk sheep, between 30 and 56 days of age, even higher differences in average gains were noted (+62–90 g). 

In the case of the second polymorphic site (*JAK2*/e6/seq) in both breeds, sheep with the *GG* genotype had the greatest body weights on day 2 (+0.4–0.5 kg), day 30 (+0.6–1 kg; only Suffolk), and 56 days of age (Pomeranian +1.5–1.7 kg and Suffolk +2.2–2.7 kg) and daily gains for the period from 2 to 56 days of age (Pomeranian +19–24 g and Suffolk +32–41 g) and for the period from 30 to 56 days of age (+60–63 g; Suffolk only) (*p* ≤ 0.01) compared to heterozygotes and *AA* genotypes. For the Suffolk breed, the weight of sheep with the *GG* genotype was also higher by 1.8 kg during mating than in sheep with the other genotypes (*p* ≤ 0.01). 

Polymorphism within exon 24 (*JAK2*/e24/*Hpy*188III) was also characterized by highly significant associations (*p* ≤ 0.01), where individuals with the *AA* genotype had the highest body weights compared to heterozygotes and individuals with the *GG* genotype on the 2nd day of life (Pomeranian +0.3–0.6 kg and Suffolk +0.4–0.8 kg) as well as on day 30 (+1.1–2.7 kg; Suffolk only) and on day 56 (Pomeranian +1.4–2.4 kg and Suffolk 2.5–5 kg). It was similar with daily gains both in the period from 2 to 56 days (Pomeranian +21–35 g and Suffolk +40–78 g) and in the period from 30 to 56 days (+54–89 g; Suffolk breed only). Also, during mating, sheep with the *AA* genotype were characterized by a higher weight than the other individuals (+1.4–5.2 kg; only in the Suffolk breed).

### 3.7. Ovine JAK2 Associations-Reproductive Traits

Table 8 presents the values of reproduction traits for Pomeranian and Suffolk sheep as well as lamb rearing in relation to three polymorphic sites located in the *JAK2* sheep gene. Genetic origin differentiated the average values of all the analyzed reproductive characteristics of mothers. In the case of the polymorphism in exon 6 (*JAK2*/*Ear*I), the highest fertility was shown by heterozygous Pomeranian sheep (95.8%), while the lowest (82.34%) was found in sheep with the *GG* genotype (82.34%). Statistically significant differences (*p* ≤ 0.01) were noted between individuals with the *GG* genotype and individuals that were homozygous *AA* (+11.96%) and heterozygous *AG* (+13.54%). In the case of the Suffolk breed, the highest fertility rate was shown in heterozygous individuals (94.41%) and the lowest in *AA* individuals (91.03%). However, no statistically significant differences were found in Suffolks. The prolificacy was similar among the genotypes within the breeds; however, statistically significant differences were noted in the Suffolk breed, where individuals with the *GG* genotype were characterized by the lowest prolificacy in comparison to individuals with the *AA* genotype (*p* ≤ 0.01) and heterozygotes (*p* ≤ 0.05). 

Regardless of breed, the best lamb survival rates were noted for heterozygous sheep (94.28% Pomeranian; 96.34% Suffolk). The lowest lamb survival rates in sheep of both breeds were found in *GG* individuals, which was confirmed statistically (*p* ≤ 0.01).

For the second polymorphic site (*JAK2*/e6/seq), there were clear differences between breeds. In Pomeranian sheep, the *GG* genotype was associated with the highest fertility (+7.52% *AG*, *p* ≤ 0.05; +9.53% *AA*, *p* ≤ 0.01), while in Suffolk sheep, similar fertility was observed in individuals with *AA* and *AG* genotypes, which significantly (*p* ≤ 0.05) differed from mothers with the *GG* genotype (+4.28–4.4%). 

The prolificacy of sheep in both groups, regardless of the genotype, did not exceed 1.18 and was similar. No statistically significant differences were observed. The Pomeranian sheep with the *GG* genotype reared the largest percentage of lambs (97.68%) compared to other mothers, and statistical differences were shown only for mothers with the *AA* genotype (+11.5%; *p* ≤ 0.01). Different relationships were confirmed statistically (*p* ≤ 0.01) in Suffolk sheep; the lamb survival was higher (92.09%) when ewes had the AG genotype than when the ewes had the GG genotype (difference +11.27%).

The relationship of the *JAK2*/e24/*Hpy188*III genotypes with fertility and lamb survival showed that, regardless of breed, sheep with *AA* genotype were characterized by the highest values of the analyzed traits (*p* ≤ 0.01; *p* ≤ 0.05). The difference in fertility between mothers with the *AA* genotype and individuals with the *GG* genotype was nearly 14% in Pomeranian sheep and 6% in Suffolk sheep, while the differences in lamb survival were approximately 12% for Pomeranian sheep and 19.5% for Suffolk sheep. In the case of the prolificacy, statistical differences were demonstrated in the Pomeranian breed, where individuals with the *GG* genotype differed significantly from the other individuals (0.14–0.19; *p* ≤ 0.01).

## 4. Discussion

Selection based on genetic markers focuses primarily on improving the performance characteristics of animals, such as milk yield and growth performance. The rate of genetic improvement can be increased by genotype-based selection [20]. Fertility traits are also increasingly taken into consideration during animal selection, as these traits directly influence production efficiency. The intensity of selection depends on factors such as the number of calves or lambs born per year and the interval between generations. In a production system that involves both milk and meat production, irregular reproduction patterns, such as long generation interval, can lead to economic unprofitability [21]. The use of SNP markers offers the possibility of selecting the appropriate genotype/haplotype or a combination of genotypes with a related phenotype, resulting in significant breeding benefits. An example of this is found in dairy and beef cattle-breeding programs in North America and Europe, where the use of molecular genetics contributes to the profitability of farms and the continuous improvement of production values [22].

### 4.1. GWAS Studies

Most of the target performance traits in livestock are polygenic, which are primarily suitable for testing using the genome-wide association study (GWAS) method [23]. Keogh et al. [24] performed GWAS (Charolaise and Limousin breeds) for SNPs involved in the reproductive potential of these breeds (calving interval, calving difficulty, and calf mortality) and selected production-related traits. The highest signals for carcass weight were noted for BTA2 (dominant effect of the myostatin gene) and BTA6 (non-SMC condensin I complex, subunit G (*NCAPG*)/ligand-dependent nuclear receptor corepressor-like (*LCORL*) locus). The ovine *JAK2* gene, located on the *Ovis aries* chromosome 2 (similarly to the ovine myostatin gene), does not appear directly as a strong signal in GWAS studies in sheep. In contrast, one region on OAR6 (13 SNPs) was associated with body weight (*NCAPG*/*LCORL* locus, similar to cattle on BTA6) [25]. An important methodological limitation of GWAS studies is the inclusion in the analyses of only frequent gene variants, i.e., SNPs that occur in more than 5% of individuals in the population, and indicating only strong signals above the conventional threshold line. Weaker signals that may be missed by GWAS analysis can be identified and described using traditional quantitative trait locus (QTL) mapping, as long as these signals are associated with genes involved in complex biological pathways and processes [26].

### 4.2. Physiological Implications

Body weight is the most crucial indicator of growth and development in cow and sheep production. It directly and indirectly affects meat and wool production as well as the reproduction of animals [27]. Growth regulation is controlled by multiple signaling pathways that change throughout the life of the animal. Doubtlessly, the somatotropic axis (GH/IGF-I along with their receptors and many signaling proteins) is involved in these complex signaling pathways. This pathway is also essential for cell growth, differentiation, and development, in particular for the modulation and amplification of gonadotropin, follicle-stimulating hormone (FSH), and luteinizing hormone (LH) activity during follicle growth in the ovary [28]. After birth, skeletal muscle development is controlled by the classical Wnt (portmanteau word formed by combining “Wingless” and “Integration-1”) signaling pathway, while the nonclassical Wnt signaling pathway mainly mediates the self-renewal and the growth of muscle fibers [29]. There is also evidence that the JAK2/STAT3 pathway mediates the regulation of muscle satellite cell differentiation [30] and regulates the expression of genes related to skeletal muscle development and energy metabolism, especially affecting the expression of the *MyoD* and *Myf5* genes [31]. The IL-6/JAK2/STAT3 pathway may be a principal mediator in denervated skeletal muscle atrophy [32]. The JAK2-STAT5 pathway is activated by the pulsatile release of GH in response to acute aerobic exercise in human skeletal muscle [33]. Even a single amino acid variation in the JAK2 molecule can carry severe consequences. In humans, most patients with polycythemia vera (PV), one-half of the cases of essential thrombocythemia (ET), and other myeloproliferative neoplasms (MPNs) cases possess an JAK2 p.V617F mutation (exon 12) [34] leading to constitutive activity of the JAK2 gene-encoded tyrosine kinase. Identical mutations of the *JAK2* gene and consequences occur in dogs [35]. Of the five SNPs analyzed in this manuscript, only one (rs210148032 in exon 16 of the bovine *JAK2* gene) may have analogous physiological implications. However, there is a lack of research in this area. A more in-depth analysis is necessary in relation to genetic implications.

### 4.3. Genetic Implications

A total of five polymorphic sites (two in cattle and three in sheep) located in different regions of the *JAK2* gene were analyzed. To our knowledge, this is the first attempt to assess the relationship between these SNPs and selected cattle and sheep production traits. They are an example of two different types of gene polymorphism, which may determine their different partial influence on the observed differences in shaping growth performance and reproductive characteristics in these animal species.

The polymorphic site rs210148032 in exon 16 of the bovine *JAK2* gene is a typical example of a missense mutation (the first letter of the codon). This determines the replacement of the isoleucine amino acid with valine at position 704 of the amino acid chain in the JH2 domain (i.e., pseudokinase) of active JAK2. Such a change may therefore determine the function of JAK2, as the JH2 domain directly regulates the activity of the tyrosine kinase JH1 domain. The rate at which amino acids are seen to mutate to other residues in homologous proteins has been extensively studied [36]. The isoleucine–valine substitution (as well as leucine–valine and leucine–isoleucine) is quite common in nature. The severity of such amino acid substitution is estimated at the levels 2–4 (http://www.insilicase.com/Web/SubstitutionScore.aspx) (accessed on 14 May 2023), where, for example, tryptophan is highly conserved and rarely changes to any other amino acid (level 17). Alanine, valine, isoleucine, and leucine are closely related in physico-chemical properties because they are small, hydrophobic residues similar to methionine [37]. Being hydrophobic, isoleucine/valine prefers to be buried in protein hydrophobic cores. This may partly explain the fact that ambiguous results were obtained in the three cattle breeds. Hereford and Limousin cows with the p.Ile704 were always the heaviest and had the best weight gains. However, Angus cows with the rare p.Val704 had higher weights, although for most traits, this was not statistically confirmed. Rather, this SNP may be linked to other SNPs of greater importance, and the significance of the change of these particular A/G nucleotides can be explained in a similar way as for silent mutations.

The second polymorphic site in exon 23 of the bovine *JAK2* gene (rs211067160) and both SNPs in exon 6 (rs160146162 and rs160146160) as well as in exon 24 (rs160146116) of the sheep *JAK2* gene are examples of silent mutations. Generally, those SNPs do not result in alteration of gene expression, and there are no changes in the function and structure of the mature protein. However, even a simple A/G change may result in posttranscriptional changes such as alterations in mRNA splicing and stability, followed by nucleocytoplasmic export through the nuclear pore complex (NPC) by receptor-mediated and Ran-GTP gradient-dependent active transport [38]. In addition, synonymous codons may influence ribosome occupancy time and thus can increase or decrease elongation rates during translation [39]. The location of the *JAK2*/e23/*Hae*III polymorphism is extremely important (codon for p.Pro1057), as it is the central part encoding the JH1 tyrosine kinase domain with the conserved tyrosine sites in JAK2: p.Tyr1007 and p.Tyr1008. Similarly, in sheep, a p.Leu1082 residue also in the JH1 domain (encoded, among others, by exon 24 where *JAK2*/*Hpy*188III SNP is positioned) may also play a crucial role in the function of kinase. The stability of this fragment determines the ability to transmit a signal inside the target cell after activation of the receptor–ligand, which is usually a member of the gp^130^ receptor family and class II cytokine–receptor family (such as interleukin 3 receptor family, erythropoietin receptor (EPO), growth hormone (GH) receptor, prolactin receptor, and thrombopoietin (TPO) receptor) [40]. Another silent A/G mutation (rs110298451) located in exon 20 of the bovine *JAK2* gene was described by Padzik and Szewczuk [17] at the third nucleotide of the lysine codon (AAA → AAG) at position 912 (p.K912) of the mature amino acid chain (also JH1 domain with tyrosine kinase activity). The influence of the polymorphism on growth traits varied depending on the breed: for the Hereford breed, the *GG* genotype was favored; for the Angus breed, the heterozygous genotype; and for the Limousin breed, the *AA* genotype. Therefore, synonymous SNP within exons encoding the JH1 kinase domain may be associated with other causative mutations in the *JAK2* gene or a completely different gene involved in developing growth performance traits in cattle. In sheep, two SNPs located in exon 6 (codons for p.Glu177 and p.Thr196) were analyzed. This exon encodes a fairly conservative FERM domain, which primarily determines the binding of the JAK2 molecule to the target receptor (receptor association). However, apart from binding to the receptor, this domain plays a role in the overall organization of the tyrosine kinase structure and its subsequent activation/deactivation: a third level of regulation of kinase activity [41]. Sheep of both breeds with a combination of *AA*/*GG/AA* genotypes of SNPs located in exon 6 and exon 24 appear to be heavier and have better daily gains compared to other individuals. Since these are the first association studies for these SNPs, the results cannot be compared to studies by other authors. Therefore, this observation needs to be confirmed statistically in separate analyses considering haplotypes or genotype combinations.

### 4.4. Involvement of JAK2 in the Formation of Reproductive Traits

The optimal level of reproduction in beef cattle herds is closely tied to the potential economic benefits, which largely rely on beef production [42,43]. Two of the important indicators determining the fertility of cows are the age of the first calving and the length of the calving interval. Earlier age at first calving reduces the cost of rearing heifers through earlier conception, which is influenced by body weight and condition of the cows [44], while the extension of the rearing period of heifers increases costs incurred by breeders [45,46]. 

Regardless of the polymorphic location, Hereford cows of the *GG* genotype calved the earliest and had the shortest calving interval compared to the *AA* genotype. In Angus cows, statistically significant differences were observed only for the *JAK2*/e16/*Rsa*I polymorphism and the calving interval, where, once more, individuals with the *GG* genotype had a shorter calving interval. Cows with a short birth intervals are usually characterized by the best fertility and the highest reproductive efficiency [45]. The length of the period in question is more important in the case of dairy cattle, while in the case of beef production, attention is paid to the calving rhythm, which means that breeders keep cows with regular calving intervals for further breeding [45]. The results obtained in this paper are difficult to relate to the studies of other authors in the context of the analyzed polymorphisms and the discussed reproductive traits. The only work on a similar subject is the study by Padzik and Szewczuk [17], which identified a silent mutation in exon 20 (dbSNP ID: rs110298451) in the *JAK2* gene in the same breeds of cattle. The age of cows at first calving was later (average days: Limousin: 1118, Hereford 1038, and Angus 1085, respectively). Moreover, the authors did not find a significant relationship between *JAK2*/e20/*Rsa*I polymorphism and age at first calving.

Reproductive traits in sheep have low heritability. Therefore, traditional selection by phenotype results in small annual genetic progress [47]. Identification of candidate genes is one strategy for improving these traits. These genes directly or indirectly affect fertility [48,49], prolificacy [50,51], and lamb rearing [51,52], which are of great importance and may contribute to increasing the rate of genetic improvement of these traits. It should be remembered that polygenic traits are traits that are constantly dispersed, referring to the existence of many genes that help in the expression of various gene traits (it is selective). Environmental elements, including animal stress, also significantly affect gene expression [27,53]. Therefore, in our study, the observations were made on the same farm, with the same management and feeding system for both sheep breeds. The sheep spent most of the year on pasture, where the nutritional properties of forage change depending on the season, which may affect gene expression. Metzler-Zebeli et al. [54] stated that different feeding levels can alter gene expression. Sheep grazing, stimulated by hot summers and cold winters, can also alter gene expression [55].

Because no reports are available in the literature on the relationships between the three polymorphic sites analyzed in our study and performance and reproduction traits of sheep, it is not possible to contrast and compare our results with those of other authors. Therefore, we discussed our results in terms of their economic importance and their influence on the genetic improvement of sheep.

In the case of polymorphisms located in exons 6 (*JAK2*/*Ear*I) and 24 (*JAK2*/*Hpy*188III) in the group of sheep with the *GG* genotype, a certain tendency was observed associated with the deterioration of fertility and lower rearing of lambs, which is also associated with the economic aspect of production [56,57].

An inverse relationship was found in sheep of the autochthonous Pomeranian breed in the case of the *JAK2*/e6/seq polymorphism, where the fertility and lamb survival in ewes of the *GG* genotype exceeded 97%, which indicates excellent reproduction. Suffolk ewes with the *GG* genotype were also characterized by high fertility (90.34%). It is assumed that the value of the indicator can be considered satisfactory when it is 90%, good if it exceeds 95%, and very good when it is close to 100% [58]. Preliminary analyses showed that ewes with the AA and AG genotypes had better fertility and lamb survival, apart from Pomeranian sheep in the case of the *JAK2*/e6/seq polymorphic site. Irrespective of the analyzed polymorphic site, the *GG* ewes reared fewer lambs than other mothers (except for sheep of the Pomeranian breed *JAK2*/e6/seq). The values of the index oscillate between 73.15–84.52%, which means that the losses of lambs in the analyzed groups of mothers with *GG* genotypes exceeded the acceptable level of 5% [59,60,61]. In these groups of sheep, there were lambs from twin pregnancies, which, according to Clune et al. [59], may be less developed and quite often show reduced viability immediately after birth [60], which delays or even prevents proper colostrum intake [62]. As a result, deaths may occur in the first hours of life. Losses during the rearing period may also be due to the lack of colostrum or its poorer quality and insufficient milk yield of mothers. For this reason, lambs are often fed with preparations containing powdered colostrum and, in further rearing, with milk replacers [61]. Therefore, it should be considered that greater care provided by breeders may improve lamb-rearing results in this breed of sheep. The selection and elimination of individuals with the *GG JAK2*/e6/seq genotype should also be considered.

Lamb survival is an important element of sheep production closely related to the prolificacy and profitability of production. The prolificacy of sheep in both breed groups was similar, regardless of the genotype, and did not exceed 1.18. The prolificacy of Suffolk sheep was relatively low (1.06–1.19). The results for PZO indicate it has good potential in terms of prolificacy [63]. 

The assumed profitability of the production of lambs for slaughter occurs from at least 1.5 lambs raised from the mother [64], which in practice is difficult in the case of native breeds, such as Pomeranian sheep, except for prolific breeds. This trait in sheep depends to a large extent on, e.g., the breed, nutrition, and age of the ewes [57,64]. Taking into account the possibilities and predispositions of the analyzed sheep breeds in the context of higher prolificacy, attention should be paid to the nutrition of ewes before and during breeding [65], which, according to Łozicki [66], could increase the number of maturing ova and, consequently, improve fertility by 10–30%. In addition, the introduction of biotechnological methods in reproduction, namely the stimulation of the functioning of the reproductive system but also the induction of superovulation, according to Skliarov et al. [67], suggests the possibility of a significant improvement in this indicator.

## 5. Conclusions

The phenotypic characteristics of cattle and sheep are the result of a complex interaction of many genetic and environmental factors, which usually act simultaneously, and it is difficult to determine the degree of influence of each of them. Therefore, early identification of genetic features in young animals enables more effective selection management and effective breeding in the herd. Production indicators, such as, body weight, daily gains of lambs and calves at different stages of their lives, as well as reproductive indices, are important factors that are crucial for successful ruminant production and thus translate into profitability of production. Identification of genomic regions and biological pathways that contribute to the understanding of variability in body weight traits and reproductive traits is important for selection purposes. Therefore, the identification of a number of genes with a moderate and low impact on the traits discussed in the paper will allow the use of genetic-marker-assisted selection for in-breed selection in order to achieve higher body weight, daily gains, and better fertility and rearing of lambs and calves in various housing and feeding conditions. The results obtained in the study are ambiguous, and additional association studies are needed on other herds and within different breeds of cattle and sheep. For this reason, an analysis of haplotypes and/or combined genotypes of the *JAK2* gene and selected polymorphic sites located in the genes encoding the somatotropic axis should be performed.

## Figures and Tables

**Figure 1 animals-13-02470-f001:**
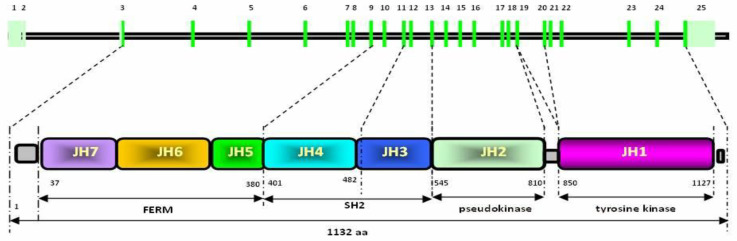
The primary structure of JAK2 kinase (own elaboration).

**Figure 2 animals-13-02470-f002:**
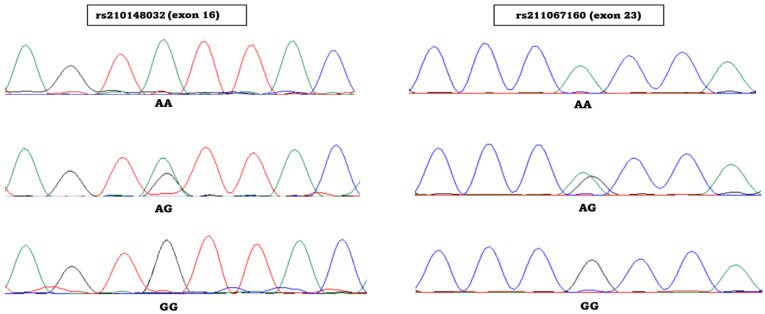
Chromatograms for two polymorphic sites identified within the bovine *JAK2* gene. green—adenine, black—guanine, red—thymine, blue—cystosine.

**Table 1 animals-13-02470-t001:** Characterization of selected single-nucleotide polymorphisms within bovine *JAK2* gene.

Primer Sequence (5′ → 3′)	Polymorphism Position ** and Type	Codon and Amino Acid(s) ***	PCR-ACRS (Restriction Enzyme, Cleavage Site, and Genotypes)
JAK2e16mF gggccctggacatactaagtg JAK2e16mR gtcttttggcaaaactgtaG *	rs210148032 g.39400906A>G Missense	ATT → GTT p.Ile704Val	*Rsa*I (gt^ac) PCR amplicon 211 bp *AA* 190 bp + 21 bp *AG* 211 bp + 190 bp + 21 bp *GG* 211 bp (no cut)
JAK2e23mF catatattgacaagagtaaaagccG * JAK2e23mR tccccacctttcaaaacttc	rs211067160 g.39383281A>G Synonymous	CCA → CCG p.Pro1057	*Hae*III (gg^cc) PCR amplicon 211 bp *AA* 211 bp (no cut) *AG* 211 bp + 185 bp + 26 bp *GG* 185 bp + 26 bp

* Mismatch is underlined (ACRS-PCR); ** position according to: RefSeq NC_037335.1 Chromosome 8 ARS-UCD1.2 Primary Assembly; *** amino acids abbreviations: Ile, isoleucine; Val, valine; Pro, proline.

**Table 2 animals-13-02470-t002:** Characterization of selected single-nucleotide polymorphisms within ovine *JAK2* gene.

Primer Sequence (5′ → 3′)	Polymorphism Position * and Type	Codon and Amino Acid **	PCR-RFLP (Restriction Enzyme, Cleavage Site, and Genotypes)
oJAK2e6F ttgaccttgttaaatgtatatgttctg	rs160146162 g.73397955A>G Synonymous	GAA → GAG p.Glu177	*Ear*I (CTCTTC(1/4)^) PCR amplicon 280 bp *AA* 280 bp (no cut) *AG* 280 bp + 170 bp + 110 bp *GG* 170 bp + 110 bp
oJAK2e6R ttgcataagaaaattacctgatagagc	rs160146160 g.73398012G>A Synonymous	ACG → ACA p.Thr196	PCR amplicon 280 bp not applicable (sequencing)
oJAK2e24F tctgcttgaaattaaatgtaccaaa oJAK2e24R tcagtgaactgcataaactgacc	rs160146116 g.73442093A>G Synonymous	CTA → CTG p.Leu1082	*Hpy*188III (TC^NNGA) PCR amplicon 261 bp *AA* 205 bp + 56 bp *AG* 205 bp + 137 bp + 68 bp + 56 bp *GG* 137 bp + 68 bp + 56 bp

* Position according to: RefSeq NC_056055.1 Chromosome 2 ARS-UI_Ramb_v2.0 Primary Assembly; ** amino acids abbreviations: Glu, glutamic acid; Thr, threonine; Leu, leucine.

**Table 3 animals-13-02470-t003:** The number and frequency of the *JAK2*/e16/*Rsa*I and the *JAK2*/e23/*Hae*III genotypes and alleles in the three bovine breeds under study.

Breed	*JAK2*/e16/*Rsa*I				Total	allele	
		*AA*	*AG*	*GG*		*A*	*G*
Hereford	*n*	161	95	20	276	0.7554	0.2446
	Frequency	0.5833	0.3442	0.0725	1.0000		
Angus	*n*	195	110	28	333	0.7508	0.2492
	Frequency	0.5856	0.3303	0.0841	1.0000		
Limousin	*n*	95	53	12	160	0.7594	0.2406
	Frequency	0.5937	0.3313	0.0750	1.0000		
Breed	*JAK2*/e23/*Hae*III				Total	allele	
		*AA*	*AG*	*GG*		*A*	*G*
Hereford	*n*	165	83	23	271	0.7620	0.2380
	Frequency	0.6088	0.3063	0.0849	1.0000		
Angus	*n*	61	132	152	345	0.3681	0.6319
	Frequency	0.1768	0.3826	0.4406	1.0000		
Limousin	*n*	28	72	43	143	0.4476	0.5524
	Frequency	0.1958	0.5035	0.3007	1.0000		

*n*, number of individuals.

**Table 4 animals-13-02470-t004:** The effect of *JAK2*/e16/*Rsa*I genotypes on the growth performance and selected reproductive traits in cows.

Breed	Genotype	Birth Weight (kg)	Average Daily Gain (g)	Weaning Weight (kg)	Age at First Calving (Days)	Body Weight at First Calving (kg)	Calving Interval (Days)
Hereford	*AA*	34.7 ^a^ (0.36)	1091 ^AB^ (13.11)	262.2 ^a^ (2.85)	1039 ^A^ (28.20)	589.4 ^Aa^ (2.11)	499 ^ab^ (16.06)
	*AG*	34.0 (0.33)	959 ^A^ (14.84)	254.3 (3.50)	975 ^a^ (31.00)	571.5 ^a^ (5.90)	420 ^a^ (13.57)
	*GG*	32.9 ^a^ (0.51)	926 ^B^ (16.60)	251.1 ^a^ (6.97)	813 ^Aa^ (37.69)	560.6 ^A^ (6.79)	399 ^b^ (16.36)
Angus	*AA*	36.2 ^a^ (0.30)	994 (7.76)	247.1 (1.87)	1076 (19.20)	564.6 (3.33)	416 ^Aa^ (2.02)
	*AG*	37.3 (0.39)	1011 (8.13)	251.6 (2.50)	1057 (26.91)	562.3 (3.81)	403 ^ab^ (2.23)
	*GG*	38.4 ^a^ (0.59)	1012 (8.58)	255.5 (7.89)	1011 (25.83)	574.5 (8.46)	389 ^Ab^ (1.22)
Limousin	*AA*	35.2 ^a^ (0.41)	1001 (23.62)	275.5 ^A^ (3.22)	1077 (14.55)	610.8 ^a^ (10.27)	469 (21.51)
	*AG*	34.2 (0.57)	1001 (35.12)	261.3 (3.61)	1045 (14.83)	610.3 ^b^ (13.77)	479 (27.96)
	*GG*	32.6 ^a^ (0.65)	954 (10.35)	248.3 ^A^ (6.19)	1090 (15.67)	581.2 ^ab^ (5.69)	466 (6.69)

The means in the columns marked with the same letters within the same breed differ significantly at: small letters, *p* ≤ 0.05; capitals, *p* ≤ 0.01. Numbers in parentheses are the standard errors of the means.

**Table 5 animals-13-02470-t005:** The effect of *JAK2*/e23/*Hae*III genotypes on the growth performance and selected reproductive traits in cows.

Breed	Genotype	Birth Weight (kg)	Average Daily Gain (g)	Weaning Weight (kg)	Age at First Calving (Days)	Body Weight at First Calving (kg)	Calving Interval (Days)
Hereford	*AA*	34.5 (0.34)	1060 ^A^ (12.86)	260.8 ^a^ (2.53)	1020 ^a^ (27.83)	584.9 (3.26)	487 ^a^ (15.87)
	*AG*	34.1 (0.38)	1018 ^B^ (24.03)	257.9 (4.95)	988 (37.85)	575.6 (4.05)	430 (14.72)
	*GG*	33.9 (1.03)	906 ^AB^ (16.51)	244.4 ^a^ (3.47)	874 ^a^ (51.07)	571.9 (7.08)	397 ^a^ (16.73)
Angus	*AA*	37.16 (0.71)	1005 (14.27)	249.6 (3.66)	1065 (25.94)	559.4 (5.42)	423 (4.37)
	*AG*	37.40 (0.42)	1017 (11.25)	250.8 (2.92)	1021 (25.59)	563.9 (4.72)	400 (1.71)
	*GG*	36.3 (0.29)	993 (7.37)	248.4 (1.99)	1084 (20.98)	566.0 (3.28)	410 (2.08)
Limousin	*AA*	35.5 (0.77)	1058 ^a^ (17.42)	274.5 ^a^ (4.38)	1075 (18.39)	580.5 ^ab^ (3.13)	413 ^a^ (28.82)
	*AG*	34.2 (0.40)	990 (32.90)	271.3 (3.91)	1078 (15.23)	618.8 ^a^ (12.95)	504 ^a^ (27.69)
	*GG*	35.2 (0.73)	967 ^a^ (36.69)	261.2 ^a^ (4.13)	1056 (25.25)	617.3 ^b^ (17.77)	483 (32.79)

The means in the columns marked with the same letters within the same breed differ significantly at: small letters, *p* ≤ 0.05; capitals, *p* ≤ 0.01. Numbers in parentheses are the standard errors of the means.

**Table 6 animals-13-02470-t006:** The number and frequency of genotypes and alleles for three different *JAK2* polymorphisms among two sheep breeds under study.

Breed	*JAK2*/e6/*Ear*I				Total	allele	
		*AA*	*AG*	*GG*		*A*	*G*
Pomeranian	*n*	32	58	48	138	0.4420	0.5580
	Frequency	0.2319	0.4203	0.3478	1.0000		
Suffolk	*n*	29	43	123	195	0.2590	0.7410
	Frequency	0.1487	0.2205	0.6308	1.0000		
Breed	*JAK2*/e6/seq				Total	allele	
		*AA*	*AG*	*GG*		*A*	*G*
Pomeranian	*n*	55	59	24	138	0.6123	0.3877
	Frequency	0.3986	0.4275	0.1739	1.0000		
Suffolk	*n*	49	110	36	195	0.5333	0.4667
	Frequency	0.2513	0.5641	0.1846	1.0000		
Breed	*JAK2/*e24*/Hpy*188III				Total	allele	
		*AA*	*AG*	*GG*		*A*	*G*
Pomeranian	*n*	43	67	28	138	0.5544	0.4456
	Frequency	0.3116	0.4855	0.2029	1.0000		
Suffolk	*n*	49	128	18	195	0.5795	0.4205
	Frequency	0.2513	0.6564	0.0923	1.0000		

*n*, number of individuals.

**Table 7 animals-13-02470-t007:** Means and standard errors (in parentheses) of weights and gains traits in sheep with different *JAK2* genotypes of three polymorphic sites.

	Breed	Genotype	Body Weight at 2 Days (kg)	Body Weight at 30 Days (kg)	Body Weight at 56 Days (kg)	Average Daily Gains between 2–56 Days (g)	Average Daily Gains between 30–56 Days (g)	Body Weight at Mating (kg)
*JAK2*/e6/*Ear*I	Pomeranian	*AA*	4.3 ^AB^ (0.04)	n/a	19.3 ^AB^ (0.19)	278 ^AB^ (3.52)	n/a	54.4 (0.43)
		*AG*	4.0 ^BC^ (0.03)	n/a	18.1 ^BC^ (0.15)	262 ^Ba^ (2.78)	n/a	53.8 (0.28)
		*GG*	3.6 ^AC^ (0.02)	n/a	17.1 ^AC^ (0.18)	251 ^Aa^ (3.20)	n/a	53.2 (0.43)
	Suffolk	*AA*	4.8 ^AB^ (0.05)	14.2 ^AB^ (0.13)	23.5 ^AB^ (0.21)	347 ^AB^ (3.44)	359 ^AB^ (7.24)	62.0 ^AB^ (0.55)
		*AG*	4.1 ^B^ (0.02)	13.0 ^B^ (0.19)	20.7 ^Ba^ (0.25)	307 ^Ba^ (4.39)	297 ^Ba^ (9.86)	58.9 ^B^ (0.39)
		*GG*	4.0 ^A^ (0.03)	12.9 ^A^ (0.15)	19.9 ^Aa^ (0.18)	294 ^Aa^ (2.96)	269 ^Aa^ (4.95)	57.9 ^A^ (0.26)
*JAK2*/e6/seq	Pomeranian	*AA*	3.8 ^A^ (0.04)	n/a	17.9 ^A^ (0.18)	261 ^A^ (3.16)	n/a	53.8 (0.31)
		*AG*	3.9 ^B^ (0.04)	n/a	17.7 ^B^ (0.16)	256 ^B^ (2.73)	n/a	54.2 (0.34)
		*GG*	4.3 ^AB^ (0.05)	n/a	19.4 ^AB^ (0.24)	280 ^AB^ (4.01)	n/a	54.9 (0.53)
	Suffolk	*AA*	4.1 ^A^ (0.06)	12.7 ^A^ (0.25)	19.8 ^A^ (0.34)	292 ^A^ (5.32)	275 ^A^ (8.56)	58.4 ^A^ (0.54)
		*AG*	4.1 ^B^ (0.03)	13.1 ^a^ (0.14)	20.3 ^B^ (0.17)	301 ^B^ (2.93)	278 ^B^ (5.79)	58.4 ^B^ (0.25)
		*GG*	4.6 ^AB^ (0.06)	13.7 ^Aa^ (0.20)	22.5 ^AB^ (0.32)	333 ^AB^ (5.29)	338 ^AB^ (8.67)	60.2 ^AB^ (0.59)
*JAK2/*e24*/Hpy*188III	Pomeranian	*AA*	4.2 ^AB^ (0.05)	n/a	19.2 ^AB^ (0.20)	279 ^AB^ (3.83)	n/a	54.4 (0.43)
		*AG*	3.9 ^AC^ (0.04)	n/a	17.8 ^AC^ (0.13)	258 ^AC^ (2.14)	n/a	53.8 (0.28)
		*GG*	3.6 ^BC^ (0.05)	n/a	16.8 ^BC^ (0.15)	244 ^BC^ (2.98)	n/a	53.2 (0.43)
	Suffolk	*AA*	4.5 ^AB^ (0.06)	14.1 ^AB^ (0.16)	22.7 ^AB^ (0.26)	338 ^AB^ (4.07)	332 ^AB^ (9.04)	60.1 ^Aa^ (0.54)
		*AG*	4.1 ^AC^ (0.02)	13.0 ^AC^ (0.12)	20.2 ^AC^ (0.15)	298 ^AC^ (2.46)	278 ^AC^ (4.99)	58.7 ^Ba^ (0.23)
		*GG*	3.7 ^BC^ (0.05)	11.4 ^BC^ (0.40)	17.7 ^BC^ (0.35)	260 ^BC^ (5.98)	243 ^BC^ (11.28)	54.9 ^AB^ (0.46)

The means in the columns marked with the same letters within the same breed differ significantly at: small letters, *p* ≤ 0.05; capitals, *p* ≤ 0.01. n/a: for the Pomeranian breed, such data were not recorded.

**Table 8 animals-13-02470-t008:** Influence of genotypes on reproductive performance of Pomeranian and Suffolk sheep breeds.

	Breed	Genotype	Fertility (%)	Prolificacy (*n*/ewe)	Lamb Survival (%)
*JAK2*/e6/*Ear*I	Pomeranian	*AA*	94.30 ^A^ (2.23)	1.15 (0.05)	93.55 ^A^ (2.21)
		*AG*	95.88 ^B^ (1.45)	1.11 (0.04)	94.28 ^B^ (1.67)
		*GG*	82.34 ^AB^ (2.70)	1.18 (0.05)	81.38 ^AB^ (3.86)
	Suffolk	*AA*	91.03 (2.89)	1.19 ^A^ (0.06)	95.40 ^A^ (1.92)
		*AG*	94.41 (1.11)	1.18 ^a^ (0.05)	96.34 ^B^ (1.48)
		*GG*	94.40 (1.66)	1.09 ^Aa^ (0.02)	84.52 ^AB^ (2.00)
*JAK2*/e6/seq	Pomeranian	*AA*	88.30 ^A^ (2.35)	1.11 (0.04)	86.20 ^A^ (2.88)
		*AG*	90.31 ^a^ (2.05)	1.18 (0.04)	89.69 (2.64)
		*GG*	97.83 ^Aa^ (1.59)	1.13 (0.05)	97.68 ^A^ (1.66)
	Suffolk	*AA*	94.62 ^a^ (1.77)	1.13 (0.04)	87.06 (2.84)
		*AG*	94.74 ^b^ (1.11)	1.14 (0.03)	92.09 ^A^ (1.53)
		*GG*	90.34 ^ab^ (2.47)	1.06 (0.03)	80.82 ^A^ (4.23)
*JAK2/*e24*/Hpy*188III	Pomeranian	*AA*	95.17 ^A^ (1.76)	1.13 ^A^ (0.04)	95.77 ^A^ (1.66)
		*AG*	93.48 ^B^ (1.79)	1.08 ^B^ (0.03)	90.66 (1.93)
		*GG*	81.33 ^AB^ (3.74)	1.27 ^AB^ (0.08)	84.00 ^A^ (4.46)
	Suffolk	*AA*	95.05 ^a^ (1.44)	1.15 (0.03)	92.64 ^A^ (2.43)
		*AG*	94.17 (1.13)	1.11 (0.03)	89.44 ^B^ (1.66)
		*GG*	88.89 ^a^ (3.81)	1.11 (0.05)	73.15 ^AB^ (5.43)

The means in the columns marked with the same letters within the same breed differ significantly at: small letters, *p* ≤ 0.05; capitals, *p* ≤ 0.01. Numbers in parentheses are the standard errors of the means.

## Data Availability

All data generated or analyzed during the study are included in this published article. The datasets used and/or analyzed in the current study are available from the corresponding author on reasonable request.

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
