# Peer review of "Association between Polymorphism in the Janus Kinase 2 (JAK2) Gene and Selected Performance Traits in Cattle and Sheep"

_animals, 2023, doi:10.3390/ani13152470_

Round 1
Reviewer 1 Report
This is an interesting manuscript aimed to perform an associative study between some polymorphisms in the Janus Kinase 2 (JAK2) gene with meat and reproductive traits in cattle and sheep. Molecular and statistical methodology is clear. However, I suggest to provide the rational to select the polymorphisms included in the study. In addition, a more clear explanation of the physiological implications of the SNPs should be useful in the discussion section. Also, I recommend considering next few minor grammar comments:
- Line 20: Which polymorphism?
- Lines 68-72: Very long sentence. I suggest dividing it in two shorter sentences.
- Line 103: Remove the year of the reference. Please make this correction through the manuscript.
- Line 107: Remove the colon sign.
- Line 116: Remove the colon sign.
- Line 122: What is the reference for the nutritional management?.
- Line 127: I suggest describing briefly the DNA extraction procedure.
- Line 192: Was the same statistical model used to analyze the categorical trait “lamb survival”?
- Line 424: Replace “and” by “with”.
- Lines 454-455: The sentence is not clear.
- Line 618: The word “lamb” is repeated.
- Line 711: Replace “Endocrinol.” by “Endocrinology”.
- Line 792: Replace “Biosci.i” by “Biosci”.
- Line 756: Replace “genomics” by “Genomics”.
- Line 865: Insert period’s signs in the abbreviated journal name.
- In discussion section I suggest to separate the long paragraphs in two shorter paragraphs.
Reviewer 2 Report
My general feeling about the research is that it is overloaded with theoretical data inconsistent with the study's particular aim.
The advantage of the study is a huge number of samples analysed.
The Introduction is concerned mainly with general knowledge of the functioning of the JAK2 gene, less about the gene’s impact on ruminants’ traits. Moreover, it does not explain why the authors have decided to analyse particular fragments of the JAK2 gene. In the Introduction, you mentioned the research results by Szewczuk 2015 (line 103) but did not mention the results by Padzik et al. (2021), which appears later in the 3.1 subsection.
Please modify the data according to the updated Ovis aries genome (please refer to https://doi.org/10.1093/gigascience/giab096)
Please, improve the Tables.
Use dots instead of commas in all tables in decimal numbers.
Table 1 is not fully presented, so I cannot read it all. The authors should also consider rearranging the tables to make them more readable. Firstly I recommend merging the primer name and 5’-3’ sequence in one column. The second column should include “polymorphism position and type”: like g.A(nucleotide position)G, missense. I recommend following the nomenclature rules in the paper DOI: 10.1111/j.1538-7836.2011.04191.x. Instead of full names of amino acids, please use the well-recognised shortcuts and add a caption to each table.
Table 2: Please, explain the meaning of R1 and R2. Remove the red bar by the name of the restriction enzyme;
in Table 4, the caption needs English improvement.
Subsection 2.7: Please provide a photo after electrophoresis separation as Supplementary material.
Subsection 2.8 : lines 1825-183 English grammar needs improvement; “official recordings” – please, provide an explanation
Subsection 3.1: how long DNA fragments have you studied, and which sequence regions do they cover (can be done in Supplement)
Lines 149-151: please move this section to the end of 2.3 subsection
Lines 172: should be “was” instead “it was “
Lines 227: “a similar results was obtained” can be removed
Line 230 – please remove “strongly”
Line 278 – “no statistical relationships for Limousine” is superfluous.
Lines 577-579: you are not consistent - you should compare mean or “from… to…” values for the breeds.
Discussion is too long. It would be best if you shortened it, especially parts in lines 457-468, 504-522, 561-566
I advise an Englih editing.
I recommend a major revision.
Lines 532-534- the sentence is misleading.
Line 179 – “is” should be “are”
Lines 182-183 – Grammarly is not correct.
